# Change of Amino Acid Residues in Idiotypic Nanobodies Enhanced the Sensitivity of Competitive Enzyme Immunoassay for Mycotoxin Ochratoxin A in Cereals

**DOI:** 10.3390/toxins12040273

**Published:** 2020-04-23

**Authors:** Caixia Zhang, Weiqi Zhang, Xiaoqian Tang, Qi Zhang, Wen Zhang, Peiwu Li

**Affiliations:** 1Oil Crops Research Institute of the Chinese Academy of Agricultural Sciences, Wuhan 430062, China; zcx1778253705@163.com (C.Z.); zhangwq-viki@outlook.com (W.Z.); wtxqtutu@163.com (X.T.); zhangwen@oilcrops.cn (W.Z.); lipeiwu@caas.cn (P.L.); 2Key Laboratory of Biology and Genetic Improvement of Oil Crops, Ministry of Agriculture, Wuhan 430062, China; 3Laboratory of Risk Assessment for Oilseeds Products, Wuhan, Ministry of Agriculture, Wuhan 430062, China; 4Key Laboratory of Detection for Mycotoxins, Ministry of Agriculture, Wuhan 430062, China; 5Quality Inspection and Test Center for Oilseeds Products, Ministry of Agriculture, Wuhan 430062, China

**Keywords:** Amino acid residue, idiotypic nanobody, immunoassay, ochratoxin a, mycotoxin

## Abstract

Anti-idiotypic nanobodies, usually expressed by gene engineering protocol, has been shown as a nontoxic coating antigen for toxic compound immunoassays. We here focused on how to increase immunoassay sensitivity by changing the nanobody’s primary sequence. In the experiments, two anti-idiotype nanobodies against monoclonal antibody 1H2, which is specific to ochratoxin A, were obtained and named as nontoxic coating antigen 1 (NCA1) and nontoxic coating antigen 2 (NCA2). Three differences between the nanobodies were discovered. First, there are six amino acid residues (AAR) of changes in the complementarity determining region (CDR), which compose the antigen-binding site. One of them locates in CDR1 (I–L), two of them in CDR2 (G–D, E–K), and three of them in CDR3 (Y–H, Y–W). Second, the affinity constant of NCA1 was tested as 1.20 × 10^8^ L mol^−1^, which is about 4 times lower than that of NCA2 (5.36 × 10^8^ L mol^−1^). Third, the sensitivity (50% inhibition concentration) of NCA1 for OTA was shown as 0.052 ng mL^−1^, which was 3.5 times lower than that of nontoxic coating antigen 2 (0.015 ng mL^−1^). The results indicate that the AAR changes in CDR of the anti-idiotypic nanobodies, from nonpolar to polar, increasing the affinity constant may enhance the immunoassay sensitivity. In addition, by using the nontoxic coating antigen 2 to substitute the routine synthetic toxic antigen, we established an eco-friendly and green enzyme-linked immunosorbent assay (ELISA) method for rapid detection of ochratoxin A in cereals. The half-maximal inhibitory concentration (IC_50_) of optimized ELISA was 0.017 ng mL^−1^ with a limit of detection (LOD) of 0.003 ng mL^−1^. The optimized immunoassay showed that the average recoveries of spiked corn, rice, and wheat were between 80% and 114.8%, with the relative standard deviation (RSD) ranging from 3.1–12.3%. Therefore, we provided not only basic knowledge on how to improve the structure of anti-idiotype nanobody for increasing assay sensitivity, but also an available eco-friendly ELISA for ochratoxin A in cereals.

## 1. Introduction

Immunoassays, such as ELISA, are a critical part of mycotoxin detection methods. Based on the reaction between the antigen and the antibody, an immunoassay was used as a quantitative or qualitative analysis of target analytes including protein, enzyme, and some small molecules. Immunoassays not only possess an excellent specificity and sensitivity, but also have advantages in having a relatively short detection time and being cost-effective [1]. Thus, many kinds of immunoassays have been developed for mycotoxin detection. Taking ochratoxin A (OTA) as an example, Zhu et al. developed an ELISA method by using the conjugation of botryoid-shaped Au/Ag nanoparticles and second antibodies connected with horseradish peroxidase (HRP) [2]. Hu et al. developed a silver nanoparticle-based fluorescence-quenching lateral flow immunoassay based on the monoclonal antibody [3]. Tang et al. developed a noncompetitive and homogeneous fluorescence resonance energy transfer immunoassay using nanobodies [4]. All the methods above exhibited satisfied sensitivity for OTA detection.

In the procedure of most mycotoxin detection methods, researchers were inevitably exposed to toxic synthetic antigens, such as the conjugation of OTA and albumin from bovine serum (OTA-BSA), even if there were effective protection measures. Under long-time exposure, damage to the researchers’ health occurs to some extent. In addition, the common synthesis procedure of these antigens demands a large amount of mycotoxin standards and organic reagents, which are also deleterious to operators and the environment [5]. Thus, the development of eco-friendly and green immunoassay methods for mycotoxin detection without using toxic synthetic antigens would be a promising field.

A great idea satisfying this demand is to develop anti-idiotype antibodies to surrogate toxic synthetic antigens used in usual immunome analysis methods. There are many kinds of anti-idiotype antibodies that have been developed. Hsu et al. developed an anti-idiotype polyclonal antibody against aflatoxin B_1_ monoclonal [6]. Guan et al. developed an anti-idiotype polyclonal antibody against aflatoxin M_1_ monoclonal [7]. Nogami et al. developed an anti-idiotype monoclonal antibody against emicizumab [8]. Wang et al. developed anti-idiotype nanobody against aflatoxin B_1_ monoclonal [5]. Among all kinds of anti-idiotype antibodies, a nanobody has the advantages of a compact structure, great stability, and solubility [9]. In addition, nanobodies can be produced through bacterial or fungicidal systems in a short period of time compared to using animals [10].

As is well known, the sensitivity of ELISA decreases with the high affinity of synthetic antigen. The high affinity of antigens inhibits antibodies and detection targets to compete for binding sites of the antigen. However, the relationship between affinity and sensitivity of protein coating antigens remains uncertain. In recent years, the construction and function of nanobodies have been gradually revealed, which provides the opportunity for us to obtain nanobodies with a different affinity constant [11]. Early studies show unique sequence characteristics of the Variable domain of heavy chain of heavy-chain antibody (VHH) domain, including antigen-binding sites consisting of three Complementarity determining region (CDR) regions [12]. The CDR3 regions are longer than conventional antibodies and generally consist of 15–22 amino acids, which increase the stability of the nanobody [13]. In the FR2 region, the heavy chain variable region of the conventional antibody comprises of four conserved fat-soluble amino acids (V, L, W, and G). However, these amino acids are mutated to the hydrophilic amino acids, such as F, E, R, and F/G, which improves the solubility of the heavy chain antibody [14].

We aimed to develop an anti-idiotype nanobody for OTA detection to deal with the problem of the serious toxicity and the broad occurrence of OTA. Ochratoxin A, a secondary fungal metabolite, is mainly produced by several *Aspergillus* species and *Penicillium* species [15]. OTA is regarded as one of the most important contaminants in food and feedstuff all over the world. Usually, the occurrence of OTA is reported in coffee, grape, soybean, bear, nuts, meat products, and nearly all kinds of cereals [16,17]. From Zaied’s report for OTA detection in Tunisian cereal in 2009, the average contamination of OTA ranged between 44–117 μg kg^−1^ for wheat, barley, rice, and sorghum [18]. This concentration of OTA is more than four times higher than the low maximum permitted level of OTA in food set by the European Commission, ranging between 2–10 μg kg^−1^ [19]. According to the research, OTA has a series of deleterious effects on several species of animals and human beings, including nephrotoxicity, hepatotoxicity, neurotoxicity, and teratogenic immunotoxicity [20,21,22,23,24,25]. Research shows Balkan endemic nephropathy (BEN) and urinary tract tumors may be introduced by OTA [26]. Hence, to humans, the International Agency for Research on Cancer (IARC) classified OTA as a possible carcinogen [27]. Consequently, it is of supreme importance to establish detecting approaches with high sensitivity and specificity.

In this study, we obtained two anti-idiotype nanobodies against monoclonal antibody 1H2, which is specific to OTA, that were named as nontoxic coating antigen 1 (NCA1) and nontoxic coating antigen 2 (NCA2). We are also interested in the question of the relationship between affinity and sensitivity of protein coating antigen. Therefore, we analyzed the prime structure, affinity constant, and sensitivity of the two NCAs. Through using the NCA with higher sensitivity to substitute the routine synthetic toxic antigen, we also established an eco-friendly and green ELISA method for the rapid detection of ochratoxin A in cereals.

## 2. Results and Discussion

### 2.1. Bio-Panning

The phage-displayed nanobodies were isolated for competitive elution by reducing the concentration of OTA standard (100, 50, 5 ng mL^−1^) gradually. The number of phage output per round of panning is shown in Appendix A. With the increase of the selection pressure in each round, the phage output increased gradually after each round of panning, which means the phage clones specific to the mAb 1H2 were effectively enriched.

### 2.2. Identification of Positive Clone

We randomly selected 30 clones from the third round of panning and their ability of binding to OTA was determined by phage ELISA. From Figure 1, it can been seen that all 30 phage clones showed inhibition of binding to antibodies by free OTA. Eleven of these 30 clones exhibited strong positive inhibition. This result indicated anti-idiotype nanobody phage with a great affinity for mAb 1H2 could be efficiently obtained by the panning method.

Gene sequencing of the eleven positive clones revealed there were only two nanobody genotypes, named NCA1 and NCA2, respectively. As can be seen in Figure 2, the amino acid sequences of the two clones have characteristic amino acid substitutions in the FR2 region, including residues E, R, and G. Moreover, the framework regions in the sequences are highly conservative and the CDR3 region possesses 16 amino acids. These results are consistent with the main characteristics of the nanobody.

### 2.3. Comparison of NCAs

#### 2.3.1. Comparison of Prime Structure and Spatial Structure

The prime structure analysis of NCAs is described in Figure 2. The obvious differences between NCAs in CDR were the six amino acids in CDR regions. Specifically, one of them located in CDR1 (I–L), two of them in CDR2 (G–D, E–K), and three of them in CDR3 (Y–H, Y–W). These amino acid residues (AARs) can be concluded in aliphatic amino acids (I, L, G, V), aromatic amino acids (Y, W), the acidic amino acids (D, E), and the basic amino acids (K, H). With the changes of nonpolar amino acids to polar amino acids (G–D, Y–H), that the CDR of NCA2 was higher polar than the CDR of NCA1.

The spatial structure of NCAs was homology-modeled through the SWISS-MODEL website. The rationality of spatial structures was estimated by the Ramachandran plots and is shown in Appendix A. When comparing the spatial structure of NCAs (Table 1), the AAR changes in CDR resulted in the spatial structure changes in CDR. Furthermore, it can be speculated that the sensitivity of nanobody may have changed as the CDR composed the antigen-binding site.

#### 2.3.2. Comparison of Affinity and Sensitivity

The affinity of antibody reflects its ability to bind with the antigen. Usually, it is regarded as a valuable antibody with high affinity when its affinity constant ranges between 10^7^–10^12^ L mol^−1^. The affinity curve is described in Appendix A. Through the formulas, the affinity constant of NCA1 and NCA2 (listed in Table 1) were 1.20 × 10^8^ L mol^−1^ and 5.36 × 10^8^ L mol^−1^, respectively. These affinity constants reflect the anti-idiotypic nanobodies (NCAs) satisfy the standard of valuable antibodies. The affinity constant of NCA2 was about 4.5-fold higher than the affinity constant of NCA1. This result shows NCA2 had a stronger binding ability with mAb 1H2 compared to NCA1.

As is well known, the high affinity of synthetic antigens inhibits antibodies and detection target to compete for binding sites of antigen, which decreases the sensitivity of ELISA. However, the relationship between affinity and sensitivity of protein coating antigens remains unclear. The sensitivity of the NCA was analyzed through competitive ELISA (Appendix A). As shown in Table 1, the IC_50_ of the eco-friendly ELISA based on NCA1 was nearly 2.7-fold of the eco-friendly ELISA based on NCA2. Therefore, the eco-friendly ELISA based on NCA2 possesses higher sensitivity than eco-friendly ELISA based on NCA1. In combination, these result support the idea that the protein coating antigen with higher affinity constant presents higher sensitivity.

### 2.4. Evaluation of NCA2-Based Antigen Substitute ELISA Method

Because of the higher sensitivity of NCA2 compared to NCA1, we applied the NCA2 to substitute antigens, such as OTA-BSA, to establish an eco-friendly ELISA for ochratoxin A in cereals. We evaluated the NCA2-based antigen substitute’s ELISA performance in methanol effect, matrix effect, and cross-reactivity.

#### 2.4.1. Evaluation of Methanol Effect

By comparing the IC_50_ value obtained from the ELISA under the methanol concentration ranging from 5–20%, we optimized the methanol content in ELISA. As shown in Figure 3, in general, the IC_50_ value increased with a concentration increase. We confirmed the methanol concentration applied for ELISA at 10% with the lowest IC_50_ value was 0.017 ng mL^−1^.

#### 2.4.2. Standard Curve

By using the OTA standard solution to perform a NCA2-based antigen substitute ELISA method under optimal situation, we obtained the standard curve (Figure 4). Formula 1, obtained by nonlinear fitting, was used to calculate the OTA concentration of samples. From the standard curve, the IC_50_ of this approach was 0.017 ng mL^−1^ with the LOD of 0.003 ng mL^−1^. The linear range of standard curve was 0.01–0.51 ng mL^−1^.
y = 17.05 + (100.86 − 17.05)/[1 + (x/0.01)^0.84^](1)

The sensitivity comparison between the NCA-based eco-friendly ELISA and the previous study are reported in Table 2. According to the date reported in the previous works, the sensitivity of the NCA-based eco-friendly ELISA was at least eight times lower than the other methods.

#### 2.4.3. Evaluation of Matrix Effect

In the matrix analysis, we estimated the ELISA method in corn, rice, and wheat matrices. As shown in Figure 5, the LOD of this approach under those matrices were less than 0.001 ng mL^−1^. Moreover, there was nearly no difference in the linear range. This indicates there are few matrix effects induced by those sample matrices.

#### 2.4.4. Evaluation of Cross-Reactivity

The cross-reactivity test was conducted to estimate the specificity of this ELISA using the idiotypic nanobody to substitute the antigen. From Figure 6, there was nearly no inhibition when aflatoxin B_1_ (AFB_1_), zearalenone (ZEN) or deoxynivalenol (DON) was used as a competitive standard solution. This phenomenon indicates the specificity of this ELISA was great.

#### 2.4.5. Evaluation of Recovery

The recovery analysis was performed to estimate the stability and accuracy of the idiotypic nanobody-based antigen substitute ELISA method. The results outlined in Table 3 show that the average recovery of the ELISA was ranged 80.0–114.8%, while the standard deviation was ranging 0.3–6.3 μg kg^−1^. These results indicate the ELISA is reliable and accurate.

## 3. Conclusions

In this work, we isolated two kinds of anti-idiotypic nanobodies against mAb 1H2, which is specific to OTA. During the research, we focused on the changes of the nanobody’s primary sequence to improve its immunoassay sensitivity. We were also interested in the relationship of affinity and sensitivity of this anti-idiotypic nanobody as a protein coating antigen. By comparing the difference in the primary structure, we found there were six AAR of changes in the complementarity determining region (CDR), which composed the antigen-binding site. One of them was located in CDR1 (I–L), two of them in CDR2 (G–D, E–K), and three of them in CDR3 (Y–H, Y–W). The changes of amino acid make the CDR of NCA2 a higher polar than the CDR of NCA1. In a further test of the affinity and sensitivity, we obtained the affinity constant of NCA1 and NCA2, which were 1.20 × 10^8^ L mol^−1^ and 5.36 × 10^8^ L mol^−1^, respectively. Moreover, the IC_50_ of NCA1 and NCA2 were 0.052 ng mL^−1^ and 0.015 ng mL^−1^, respectively. These results show the NCA2 possesses a higher affinity constant with better sensitivity compared to NCA1. The results indicate that the AAR changes in CDR of the anti-idiotypic nanobodies increasing the affinity constant may enhance the immunoassay sensitivity. Based on NCA2 to substitute the routine synthetic toxic antigen, we established an eco-friendly and green ELISA method for the rapid detection of ochratoxin A in cereals. After optimization of NCA2-based ELISA, the IC_50_ was 0.017 ng mL^−1^ with a limit of detection of 0.003 ng mL^−1^. The average recoveries of spiked corn, rice, and wheat were between 80% and 114.8% with a abbreviation of relative standard deviation (RSD) of 3.1–12.3%.

In conclusion, we provided not only basic knowledge on how to improve the structure of anti-idiotype nanobody for increasing assay sensitivity, but also an available eco-friendly ELISA for ochratoxin A in cereals.

## 4. Materials and Methods 

### 4.1. Materials and Reagents

All reagents were of analytical grade unless otherwise stated. The monoclonal antibody (mAb) 1H2 against ochratoxin A was produced in our laboratory [28]. Ochratoxin A, aflatoxin B_1_, zearalenone, deoxynivalenol standard, bovine serum albumin (BSA), polyethylene glycol 8000 (PEG8000), Freund’s incomplete adjuvant, 3,3’,5,5’-Tetramethylbenzidine (TMB), and goat anti-mouse monoclonal antibody conjugated to horseradish peroxidase (HRP) were purchased from Sigma (St.Louis, MO, USA). E. coli ER2738 competent cells from the E. coli ER2736 line were purchased from Lucigen Corp. (Middleton, WI, USA). Top 10F’ competent cells and the Superscript III First-Strand Synthesis system were purchased from Life Technologies (Grand Island, NY, USA). The M13 bacteriophage antibody conjugated to HRP was purchased from GE Healthcare (Piscataway, NJ, USA). Helper phage M13KO7 and SfiI were obtained from New England Biolabs (Ipswich, MA, USA). Tween 20 was obtained from J&K Scientific (Beijing, China). xTractor buffer for protein extraction and His60 Superflow resin were purchased from Clontech Laboratories, Inc. (Mountain View, CA, USA). The QIAprep Spin MiniPrep Kit, QIAquick Gel Extraction Kit, and QIAquick PCR Purification Kit were all from Qiagen (Munich, Germany). The LeukoLOCK Total RNA Isolation System was obtained from Applied Biosystems (Foster City, CA, USA). Costar 96-well ELISA plates were purchased from Corning Incorporated (Corning, NY, USA). The pComb3X phagemid vector was a generous gift from Dr. Carlos F. Barbas (Scrips Research Institute, La Jolla, California).

Phosphate-buffered saline (PBS, 0.01 M, pH 7.4) was prepared by adding 8 g of NaCl, 2.9 g of Na_2_HPO_4_·12H_2_O, 0.2 g of KH_2_PO_4_, and 0.2 g of KCl in 1000 mL of deionized water. Phosphate-buffered saline with Tween 20 (PBST) was prepared by dissolving the Tween 20 with PBS buffer.

### 4.2. Safety

Pure mycotoxin standards were handled in a hood with extreme caution. All items coming in contact with a mycotoxin, phage, and bacterial cultures (glassware, vials, tubes, ELISA plates, etc.) were immersed in a solution of 2.5% NaClO and 0.25 mol L^−1^ NaCl for 1–2 h before they were discarded or autoclaved.

### 4.3. Alpaca Immunization

A 3-year-old castrated male alpaca was subcutaneously immunized with 200 μL of antigen solution (200 μg mAb 1H2 dissolved with 100 μL 0.01M PBS and mixed with 100 μL Freund’s incomplete adjuvant). This injection was performed every two weeks for a total of eight injections. To isolate lymphocytes, 20 mL of blood was collected one week after the fifth, sixth, seventh, and eighth injections. The lymphocytes were stored at −80 °C until used for total RNA extraction.

### 4.4. Phage-Displayed Library Construction

The total RNA in the blood of the alpaca was isolated according to the operation manual of the Life Technology LeukoLOCK Total RNA Extraction Kit. The first strand of cDNA was synthesized using the Superscript III First-Strand Synthesis system. The heavy chain antibody variable region VHH gene fragment from IgG2 and IgG3 was amplified by PCR using the primers designed previously: forward primer VHH-F (CAT GCC ATG ACT GTG GCC CAG GCG GCC CAG KTG CAG CTC GTG GAG TC) targeting framework 1 region; and reverse primers VHH-R1 (CAT GCC ATG ACT CGC GGC CGG CCT GGC CGT CTT GTG GTG TTG GTG TCT TGG G) and VHH-R2 (CAT GCC ATG ACT CGC GGC CGG CCT GGC CAT GGG GGT CTT CGC TGT GGT GCG) correspond to the IgG2 and IgG3 hinge regions, respectively [32]. The PCR product was purified by the QIAquick Gel Extraction Kit and digested with SfiI restriction enzyme. The digested VHH gene fragment was ligated to the vector pComb3X fragment in a 3:1 molar ratio by T4 ligase. The recombinant plasmids were introduced into competent cells of strain ER2738 of E. coli cells by electroporation, and a small aliquot of the electroporated cells was serially diluted and plated on LB ampicillin agar plates to determine the library size. The resulting library had and estimated the size of 9×107 independent clones. Twenty monoclonal clones were randomly picked from the plates for VHH gene sequencing to assess library diversity.

### 4.5. Bio-Panning

The solution of 100 μL mAb 1H2 (20 μg ml^−1^) in 0.01M PBS in a 96-well microtiter plate, and 100 μL 3 % BSA/PBS in wells of another microtiter plate, were incubated at 4 °C for 12 h. After being washed with 0.05% PBST three times, 100 ul of the freshly prepared phage display nanobody library was added to the wells coated with BSA to reduce BSA-bound phage, while the wells coated with mAb 1H2 was blocked with 3% milk–PBST solution. The microtiter plates were incubated for 1 h at 37 °C. Then, the blocked microtiter plate was washed with 0.05% PBST three times. The phage peptide library supernatant was then transferred to wells coated with an antibody, shaken at room temperature for 2 h, and then washed ten times with 0.05% PBST. The bound phages were competitively eluted in 100 μL of 100, 50, and 10 ng mL^−1^ OTA standard in 10% methanol/PBS in three rounds of bio-panning and shaken at room temperature for 30 min. The elution solution was then collected and used to infect E. coli ER2738 for the amplification of phage. The amplified phage was used for each subsequent round of panning. The titer of the phage of elution solution in each round represents the level of positive clone enrichment.

### 4.6. Identification of Positive Clone

After the last round of panning, 30 individual clones were randomly selected to identify positive clones by performing an indirect competitive phage ELISA. The competitive procedure was the same as the bio-planning. However, after washing ten times with 0.05% PBST, 100 μL of 0.2 μg mL^−1^ M13 bacteriophage antibody–HRP in 0.01M PBS was added into those wells. After incubating for 1 h at 37 °C, the microtiter plate was washed with 0.05% PBST for six times. Next was the addition of 100 μL color reagent (9.5 mL of 0.1 M citrate acetate buffer, pH 5.5, 0.5mL of 2 mg mL^−1^ TMB in ethanol, 32 μL of 30 mg mL^−1^ Urea hydrogen peroxide) and incubation for 15 min at 37 °C. The optical density (OD) values of those wells were obtained by a microplate reader. Clones that bound to mAb 1H2 (high OD value) but not to BSA (OD value closed to zero) were considered positive and were selected for further characterization. The pComb3X phagemid vector encoding the VHH was extracted from the ER2738 clone and sequenced using primer gback (GCCCCCTTATTAGCGTTTGCCATC).

### 4.7. Expression and Purification of Nanobody

The pComb3X phagemid vector encoding VHH having a unique DNA sequence was transformed into and expressed in a non-inhibitor E. coli strain TOP10F’ cell. For expression, 100 mL of SB medium was incubated with an overnight culture of TOP10F’ cells carrying the VHH expression plasmid, and was incubated at 37 °C with shaking at 250 rpm. When the culture OD_600_ value reached 0.6–0.8, a final concentration of 1 mM isopropyl-beta-D-thiogalactopyranoside (IPTG) solution was added, and then continued to incubate at 37 °C for 250 rpm overnight. Protein was isolated using xTractor buffer according to the manufacturer’s instructions. The nanobodies containing 6 × His tag were purified by Ni-NTA metal affinity chromatography according to the manufacturer’s instructions. The purity and size of the nanobodies were identified using 15% reduced sodium dodecyl sulfate polyacrylamide gel electrophoresis (SDS-PAGE) according to standard protocols. The concentration of nanobodies were determined by Bradford Method. For the special use in this paper, we named the nanobody as the nontoxic coating antigen (NCA).

### 4.8. Comparison of NCAs

The comparison of NCAs was conducted from the primary structure, spatial structure, affinity constant, and sensitivity of the obtained nanobodies.

#### 4.8.1. Comparison of Prime Structure and Spatial Structure

The primary structure of nanobodies was sequenced in the procedure of identification of positive clone. The spatial structure of nanobodies was homology-modeled by the SWISS-MODEL website using the amino acid sequences of nanobodies.

#### 4.8.2. Comparison of Affinity and Sensitivity

The affinity was measured according to Beatty’s method [33]. The procedures of measurement were the same as competitive ELISA. The gradient dilutions of mAb 1H2 and NCAs were performed in the ratio of two. The competitive ELISA experiments were performed in the way of the checkboard test. By fitting the relation between the concentration of mAb 1H2 and the OD value of diluted coating antigen solution, we calculated the concentration of mAb 1H2 corresponding to half of the ODmax value. The affinity of NCAs was the mean value of affinity constants obtained in three kinds of concentration ratio of NCA. Among these formulas, (Ab)t, (Ab′)t, (Ab″)t, and (Ab‴)t represented the four kinds of mAb 1H2 concentration corresponding to 1/2 ODmax. 

When the concentration ratio of coating antigen was 1:2
K = 1/2[2(Ab′)t − (Ab)t](2)
or
K = 1/2[2(Ab″)t − (Ab’)t](3)
or
K = 1/2[2(Ab‴)t − (Ab″)t](4)
When the concentration ratio of coating antigen was 1:4
K = 3/2[4(Ab″)t − (Ab)t](5)
or
K = 3/2[4(Ab‴)t − (Ab′)t](6)
When the concentration ratio of coating antigen was 1:8
K = 7/2[8(Ab‴)t − (Ab)t](7)

The sensitivity of the nanobody was measured by competitive ELISA under the same condition. The specific steps were the same as ELISA that were employed in the identification of positive clone. The IC_50_ was used to estimate the sensitivity of those two NCAs.

### 4.9. Evaluation of the NCA-Based Eco-Friendly ELISA

The NCA-based eco-friendly ELISA was developed based on competitive ELISA. The specific steps were roughly same as ELISA that were employed in the identification of positive clone. The only differences of this kind method were using NCA as a new kind of coating antigen and using mAb 1H2 as antibody. The principle of this method was NCA and OTA standard solutions competed the binding site of mAb 1H2.The optimal work concentration of NCA and mAb 1H2 was determined by performing a competitive ELISA in the checkboard test. The concentration groups corresponding to the OD value near to one were employed for further estimates by competitive ELISA.

#### 4.9.1. Evaluation of Methanol Effect

To evaluate the effects of methanol on the performance of the NCA-based eco-friendly ELISA, the OTA standard used in ELISA was diluted by 0.01M PBS in 5%, 10%, and 20%. By comparing the IC_50_ values obtained in different concentrations of methanol, we could determine the optimal methanol concentration used in ELISA.

#### 4.9.2. Evaluation of Matrix Effect

The matrix effect was analyzed through spiked samples including corn, rice, and wheat. These samples were pretreated as the procedure of sample preparation. By comparing the IC_50_ values of different performances when using the OTA standard solution diluted with three kinds of matrices, separately, we could estimate the effect of matrices to the idiotypic nanobody-based antigen substitute ELISA.

#### 4.9.3. Standard Curve

The standard curve of the NCA-based eco-friendly ELISA was obtained through competitive ELISA by using the OTA standard solution. The concentration of OTA ranged from 0 to 10 ng mL^−1^. The calculation of LOD and linear range were conducted by previous research [34].

#### 4.9.4. Evaluation of Cross-Reactivity

In the cross-reactivity test of the NCA-based eco-friendly ELISA, we employed three standard solutions including aflatoxin B_1_, Zearalenone, and deoxynivalenol as detecting targets to evaluate the specificity of the NCA-based eco-friendly ELISA.

#### 4.9.5. Evaluation of Recovery

Recovery analysis of the NCA-based eco-friendly ELISA was also conducted through spiked samples such as corn, rice, and wheat. The OTA standard (10, 20, 50 ng mL^−1^) was added into blank sample powders, separately. After extraction, the detection was performed three times on the same day to estimate the accuracy within the assay, and conducted five days later to estimate the accuracy between assay.

### 4.10. Sample Preparation

Five grams of ground corn, rice, and wheat samples were weighed, extracted with 80% aqueous methanol solution, and incubated at room temperature for 30 min with shaking at 250 rpm. Each sample extract was then filtered twice and centrifuged at 6000 g for 15 min. After dilution, the supernatant was used for ELISA analysis.

## Figures and Tables

**Figure 1 toxins-12-00273-f001:**
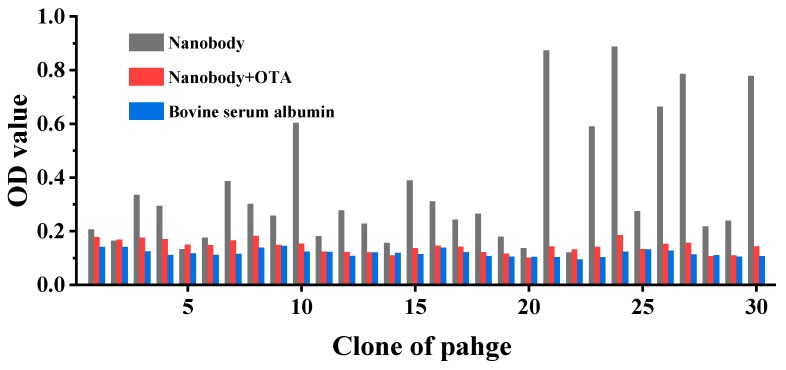
Identification of positive clones by phage-ELISA. From the optical destiny value (OD) Eleven of clones showed inhibition of binding to antibodies by free ochratoxin A (OTA).

**Figure 2 toxins-12-00273-f002:**
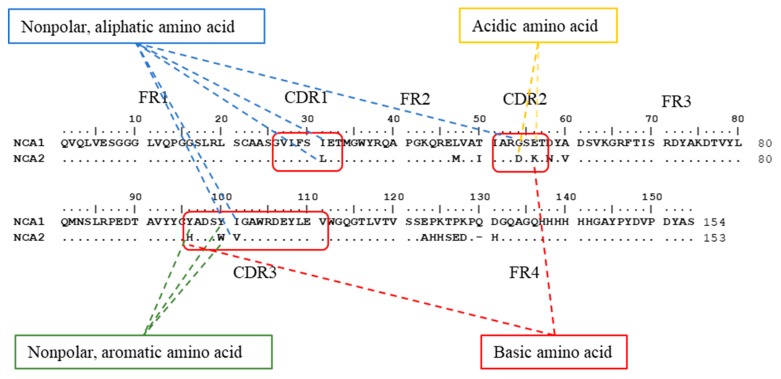
Alignment of amino acid sequences of nontoxic coating antigens (NCAs). The obvious differences between NCAs in complementarity determining region (CDR) were the six amino acids in CDR regions. Specifically, one of them located in CDR1 (I–L), two of them in CDR2 (G–D, E–K), and three of them in CDR3 (Y–H, Y–W). These amino acid residues (AAR) can be concluded in aliphatic amino acids (I, L, G, V), aromatic amino acids (Y, W), the acidic amino acids (D, E), and the basic amino acids (K, H). With the changes of nonpolar amino acids to polar amino acids (G–D, Y–H), that the CDR of NCA2 was higher polar than the CDR of NCA1.

**Figure 3 toxins-12-00273-f003:**
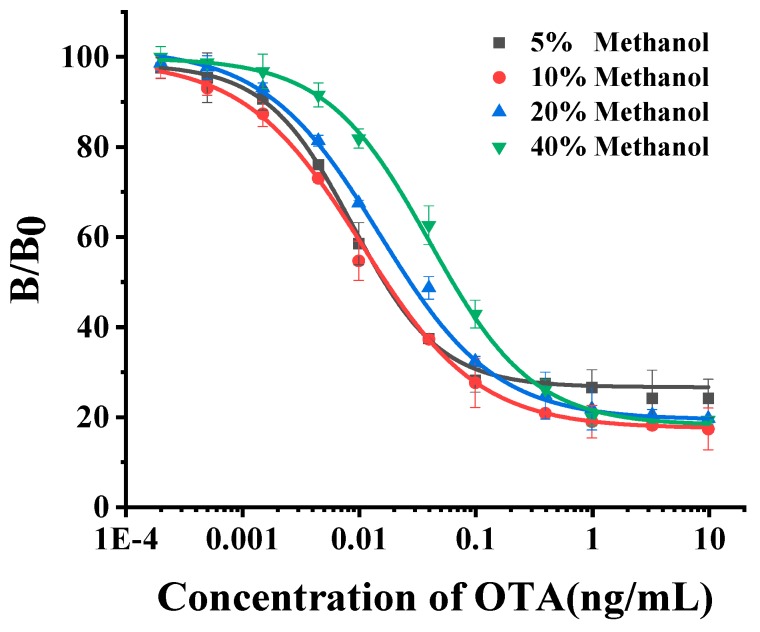
Evaluation of NCA2-based antigen substitute ELISA method under a serial concentration of methanol. When the concentration of methanol solution was 5%–40%, the IC_50_ values were 0.017, 0.017, 0.027, and 0.064 ng mL^−1^, respectively.

**Figure 4 toxins-12-00273-f004:**
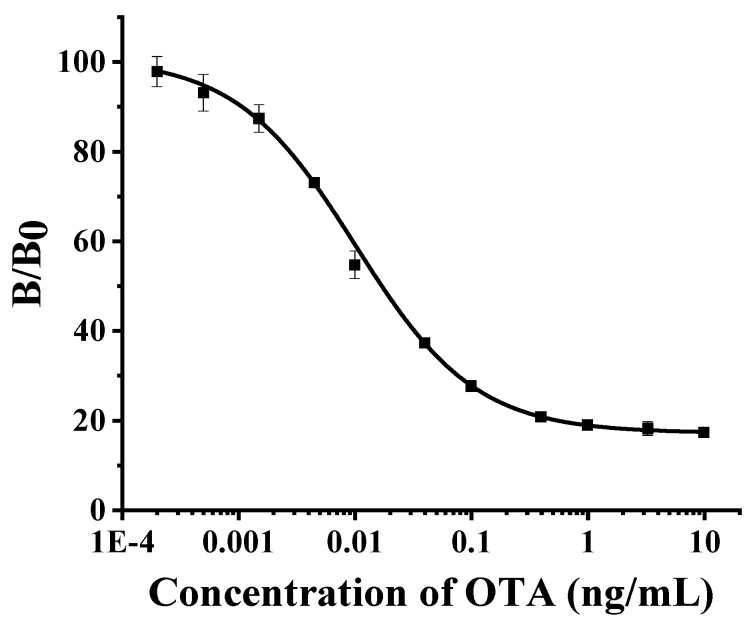
Standard curve of NCA2-based antigen substitute ELISA method. The IC_50_ of this approach was 0.017 ng mL^−1^ with the limit of detection (LOD) of 0.003 ng mL^−1^. The linear range of standard curve was 0.01–0.51 ng mL^−1^. The formula used to detection was y = 17.05 + (100.86 −17.05)/[1 + (x/0.01)^0.84^].

**Figure 5 toxins-12-00273-f005:**
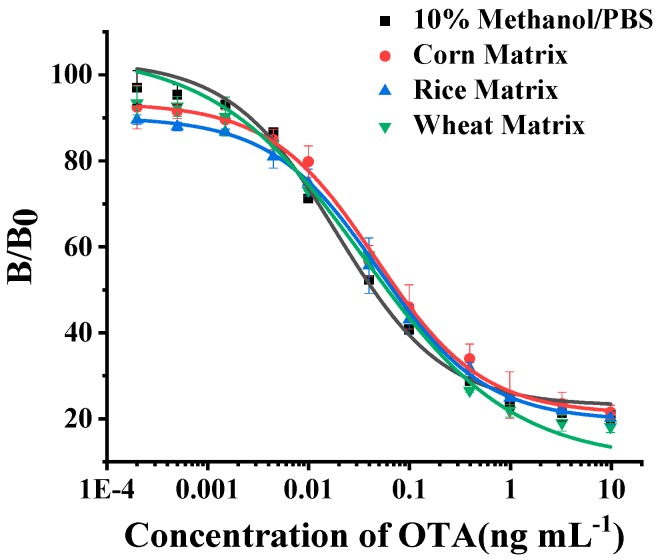
Evaluation of NCA2-based antigen substitute ELISA method under different matrices. There was little change in the sensitivity of ELISA with matrix changes. The LOD of ELISA under these matrices was 0.003, 0.004, 0.004, and 0.003 ng mL^−1^, respectively. The linear range of ELISA curve under these matrices was 0.01–0.51 ng mL^−1^, 0.01–0.62 ng mL^−1^, 0.01–0.65 ng mL^−1^, and 0.01–0.57 ng mL^−1^, respectively. PBS was the abbreviation of phosphate-buffered saline (0.01 M, pH 7.4).

**Figure 6 toxins-12-00273-f006:**
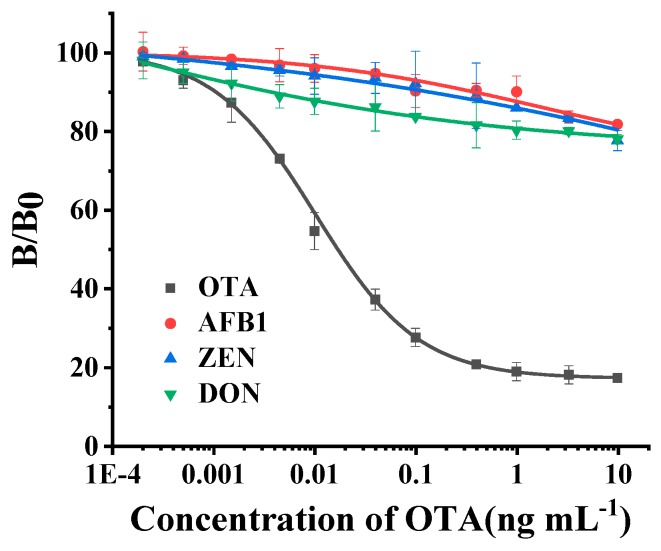
Evaluation of NCA2-based antigen substitute ELISA method in the cross-reactivity test. The IC_50_ value of ELISA could not be detected when the competitive standard solution was replaced by aflatoxin B_1_ (AFB_1_), zearalenone (ZEN) or deoxynivalenol (DON), which means there was no cross-reactivity in NCA2-based antigen substitute ELISA method.

**Table 1 toxins-12-00273-t001:** The characters of NCA1 and NCA2.

Name	Structure ^a^	Affinity Constant (L mol^−1^)	IC_50_ (ng mL^−1^)
NCA1	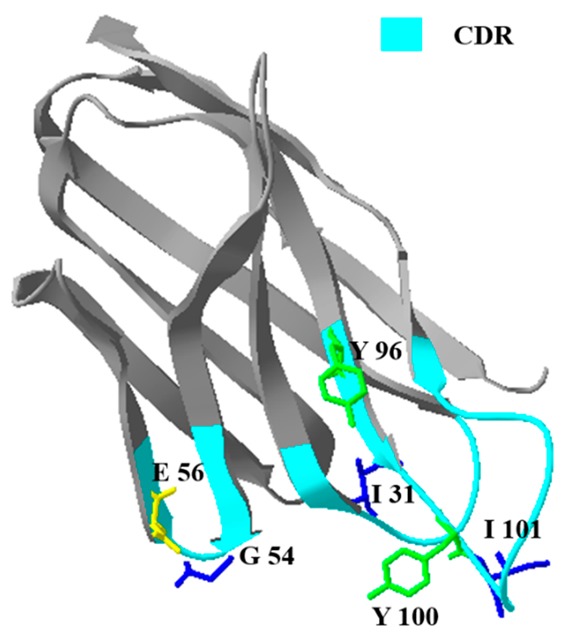	1.20 × 10^8^	0.052
NCA2	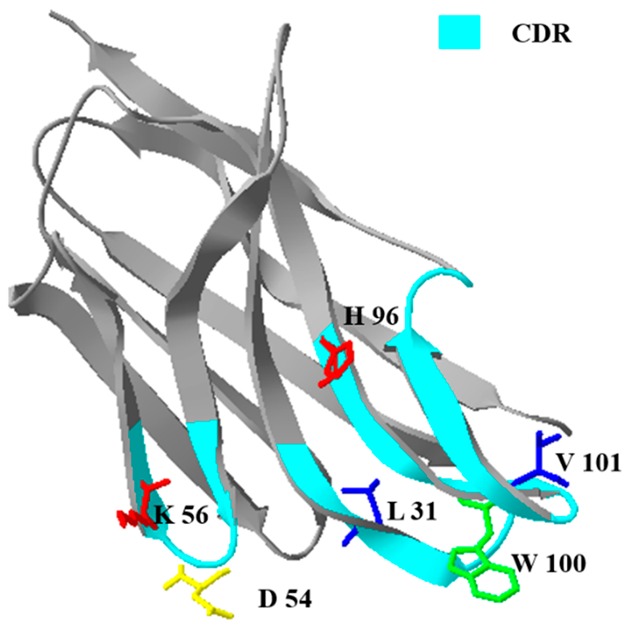	5.36 × 10^8^	0.015

^a^ the amino acid residues in red, yellow, blue, and green represent the basic amino acid, acidic amino acid, nonpolar aliphatic amino acid, and nonpolar aromatic amino acid, respectively.

**Table 2 toxins-12-00273-t002:** Sensitivity comparison between this approach and previous reports from 2010 to 2020.

Assay Methods	Assay Target	IC_50_ (ng mL^−1^)	Reference
A sensitive immunoaffinity column-linked ELISA	OTA ^1^	0.058	[28]
Botryoid-shaped Au/Ag nanoparticles enhanced ELISA	OTA	0.05	[2]
Nanobody-based ELISA	OTA	0.64	[29]
Nanobody-AviTag fusion protein-based biotin-streptavidin-amplified ELISA	OTA	0.14	[30]
CdTe quantum dots based direct and indirect competitive fluorescence-linked immunosorbent assays	OTA	0.63 and 0.234	[31]
Current method	OTA	0.017	

^1^ OTA was the abbreviation of ochratoxin A.

**Table 3 toxins-12-00273-t003:** Recovery analysis of OTA by NCA2-based ELISA.

Name	Spiked (μg kg^−1^)	Measured + SD ^1^ (μg kg^−1^)	Average Recovery (%)	RSD ^2^ (%)
**intra assay (*n* = 3) ^3^**				
Corn	50	57.4 ± 4.8	114.8	8.4
	20	17.1 ± 0.9	85.3	5.3
	10	8.0 ± 0.4	80.0	5.0
Rice	50	49.3 ± 6.0	98.6	12.2
	20	16.6 ± 0.3	83.0	1.8
	10	9.6 ± 0.8	96.0	8.3
Wheat	50	52.9 ± 5.4	105.8	10.2
	20	17.7 ± 0.6	88.5	3.4
	10	9.5 ± 0.4	95.0	4.2
**inter-assay (*n* = 5) ^4^**				
Wheat	50	51.3 ± 6.3	102.6	12.3
	20	17.9 ± 0.8	89.5	4.5
	10	9.6 ± 0.3	96.0	3.1

^1^ SD was the abbreviation of standard deviation; ^2^ RSD was the abbreviation of relative standard deviation; ^3^ experiments were carried out in three replicates on the same day; ^4^ assays were carried out on the fifth different day.

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
