# Peer review of "Change of Amino Acid Residues in Idiotypic Nanobodies Enhanced the Sensitivity of Competitive Enzyme Immunoassay for Mycotoxin Ochratoxin A in Cereals"

_toxins, 2020, doi:10.3390/toxins12040273_

Round 1

Reviewer 1 Report

The manuscript is very interesting because present an innovative results. But the paper shows a big gap about the background of immunosystem yet developed for OTA. Please, the authors must improve the background of the work yet done for the detection of OTA. For the matrix study, the authors must report in a table the working range, detection limit, IC50 and sensitivity of the immunoassay in all matrix in comparison when the assay has been carried out in buffer. Another important point is the extrapolation of the concentration of OTA in matrix using the calibration curve and the 4-logistic sigmoidal formula. Please report the four parameters obtained for all matrix in comparison when the assay has been carried out in buffer. Minor corrections: - table2: Why is the recovery from corn so high for 50 µgkg-1? Improve the written English (several tape mistake are present in the text).

Author Response

Response to Reviewer 1 Comments

First of all, thanks for your kind and effective work. On behalf of my co-authors, thank you very much for giving us an opportunity to revise our manuscript, we appreciate you very much for the positive and constructive comments and suggestions on our manuscript. The response to your comments and advice were listed as follows:

Point 1: the paper shows a big gap about the background of immune system yet developed for OTA. Please, the authors must improve the background of the work yet done for the detection of OTA.

Response 1: Thank you very much for your constructive suggestion. We have supplied the background of OTA detection method in the manuscript. such as follow sentences:

The dominant mycotoxin detection methods were chromatography and immunoassay up till now. The principle of chromatography was taking advantage of the partition coefficient of components between the stationary phase and the mobile phase to analyze mycotoxins. The typical chromatography included thin-layer chromatography, high-performance liquid chromatography, and high-performance liquid chromatography-mass spectrometry. These approaches possess outstanding sensitivity and high-throughput. But, the use of these methods was limited by the need of expensive instruments and skill people.

Point 2: For the matrix study, the authors must report in a table the working range, detection limit, IC50 and sensitivity of the immunoassay in all matrix in comparison when the assay has been carried out in buffer.

Response 2: Thank you very much for your kind suggestion. We have added these data into the manuscript (line 177 to 179). According to these data, we could come to a conclusion there were little matrix effects in this method.

IC50

(ng/mL)

LOD

(ng/mL)

Linear range

(ng/mL)

Buffer

0.044

0.03

0.01-0.51

Corn matrix

0.076

0.04

0.01-0.62

Rice matrix

0.069

0.04

0.01-0.65

Wheat matrix

0.061

0.03

0.01-0.57

Point 3: Another important point is the extrapolation of the concentration of OTA in matrix using the calibration curve and the 4-logistic sigmoidal formula. Please report the four parameters obtained for all matrix in comparison when the assay has been carried out in buffer.

Response 3: Thank you very much for your detailed advice. The calculation of OTA in real samples was using the standard curve obtained by OTA standard solution. This calculation method was based on there was little matrix effects in this method, which have been proved in Figure 4.

Point 4: Minor corrections: - table 2: Why is the recovery from corn so high for 50 µg kg-1?

Response 4: Thank you very much for your detailed advice. The concentration range of OTA was chosen according to the food and feedstuff regulation of China, GB 2761-2017 and GB 13078.2-2006. In the regulations, the limit of OTA concentration was from 2 to 100 µg/kg.

Point 5: Improve the written English (several tape mistakes are present in the text).

Response 5: Thank you very much for your constructive suggestion, we have checked the whole manuscript and refer to a native English speaker to rephrase some expression of this manuscript.

At last, allow me to thank you again for your careful advice for our manuscript!

Reviewer 2 Report

It is a well written paper regarding the change of amino acid residues in idiotypic nanobodies that enhanced the sensitivity of competitive enzyme immunoassay for mycotoxin ochratoxin A in cereals.

No comments.

Author Response

Thank you very much, we have checked and corrected some informal English expression of the manuscript.